# 🍉 SUICA: Learning Super-high Dimensional Sparse Implicit Neural Representations for Spatial Transcriptomics

Qingtian Zhu [* 1]  Yumin Zheng [* 2 3]  Yuling Sang [4]  Yifan Zhan [1]  Ziyan Zhu [5]  Jun Ding [2 3 6]  Yinqiang Zheng [1]

## Abstract

Spatial Transcriptomics (ST) is a method that captures gene expression profiles aligned with spatial coordinates. The discrete spatial distribution and the super-high dimensional sequencing results make ST data challenging to be modeled effectively. In this paper, we manage to model ST in a continuous and compact manner by the proposed tool, SUICA, empowered by the great approximation capability of Implicit Neural Representations (INRs) that can enhance both the spatial density and the gene expression. Concretely within the proposed SUICA, we incorporate a graph-augmented Autoencoder to effectively model the context information of the unstructured spots and provide informative embeddings that are structure-aware for spatial mapping. We also tackle the extremely skewed distribution in a regression-by-classification fashion and enforce classification-based loss functions for the optimization of SUICA. By extensive experiments of a wide range of common ST platforms under varying degradations, SUICA outperforms both conventional INR variants and SOTA methods regarding numerical fidelity, statistical correlation, and bio-conservation. The prediction by SUICA also showcases amplified gene signatures that enriches the bio-conservation of the raw data and benefits subsequent analysis. The code is available at https://github.com/Szym29/SUICA.

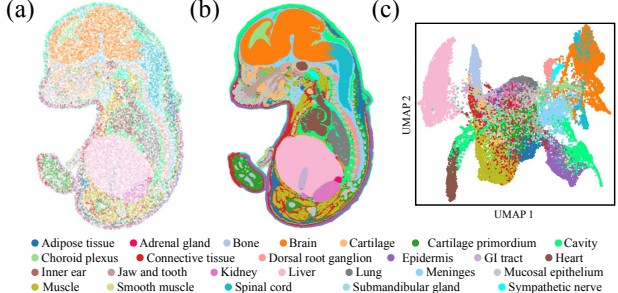

*Figure 1.* Starting with the discretely sampled spots (a) of ST, SUICA performs continuous modeling (b) by aid of the great approximation power of INRs. This approach enables complete profiling of cell heterogeneity, as visualized in the UMAP (c), which further facilitating the discovery of new biology.

## 1. Introduction

Spatial Transcriptomics (ST) enables scientists to quantify gene expression while preserving spatial information in tissue sections (Marx, 2021). These platforms use various strategies to capture mRNA transcripts from tissue sections and perform sequencing to quantify the gene expressions at spatially defined locations (Moses & Pachter, 2022). However, various degradation patterns are being observed due to some limitations throughout the data acquisition process. For instance, achieving higher resolution is essential for accurately modeling and analyzing cellular functions (Williams et al., 2022), while for now, ST data can still be rather expensive, *e.g.*, $3,500/$cm^2$ for the capture chip and $800/$cm^2$ for the high-density sequencing according to Chen et al. (2022a). Beyond the cost constraints to get high-resolution ST, another key technical challenge lies in the drop-out rates (high sparsity of ST data) that makes the gene expressions less informative for analysis (Qiu, 2020). Compounding this challenge, different sequencing techniques and platforms exhibit significant heterogeneity in terms of spatial distribution, sequencing depth (number of mRNA readouts per spot), and drop-out rates (Moses & Pachter, 2022). As a result, such the difference makes a general backbone to model and analyze data across different tissues by different platforms less feasible. To address these technical challenges, we propose a computational framework

---

*Equal contribution  [1]The University of Tokyo  [2]McGill University  [3]MUHC Research Institute  [4]Duke-NUS Medical School  [5]Carnegie Mellon University  [6]Mila - Quebec AI Institute. Correspondence to: Jun Ding <jun.ding@mcgill.ca>, Yinqiang Zheng <yqzheng@ai.u-tokyo.ac.jp>.

*Proceedings of the 42$^{nd}$ International Conference on Machine Learning*, Vancouver, Canada. PMLR 267, 2025. Copyright 2025 by the author(s).

that transforms discrete ST data into a continuous, compact representation, and thus enables complete reconstruction of ST slides by recovering the degradations in the raw data without the need to know the specific type of degradation.

Recently, Implicit Neural Representations (INRs) have drawn great attention from researchers for their compact, continuous, and differentiable properties as a novel representation of general coordinate-based signals. INRs map coordinates to corresponding values with a neural network with wide applications in inverse graphics (Mildenhall et al., 2020), geometric modeling (Park et al., 2019), and video compression (Chen et al., 2021). Motivated by INRs' inherent smoothness property, we leverage this characteristic to interpolate between ST spots, enabling more comprehensive spatial transcriptomics analysis. However, applying INRs to ST data faces two major challenges. First, scaling INRs to the super-high dimensional space of gene expression is non-trivial, as simply widening or deepening the network cannot effectively overcome the curse of dimensionality. Second, the low mRNA transcript capture rate in ST, combined with varying gene expression patterns across cellular states, results in zero-inflated ST data with high sparsity (Piwecka et al., 2023). This sparsity makes it particularly challenging for conventional INRs to accurately capture the underlying complex, non-linear spatial patterns.

Building upon this approach, we introduce SUICA, a powerful variant of INR designed especially for ST data. SUICA fully accounts for the unique properties of ST and the diversity of current mainstream ST platforms. In SUICA, we employ an Autoencoder (AE) based on Graph Convolutional Network (GCN) (Kipf & Welling, 2017) to bridge the gap between conventional INRs and ST. The GCN enhances the approximation capability for unstructured data by leveraging contextual information and making the embeddings structure-aware. By the aid of the strong profiling power of Graph Autoencoder (GAE) regarding the sparse and skewed distribution, we perform the fitting of INRs at the expressive low-dimensional embedding space, which has been proven to be compact and more suitable for INRs. To enforce the sparsity within the regression-based scheme of INRs, we construct pseudo-probabilities and adopt a regression-by-classification approach for training.

With SUICA, we are enabled to reconstruct diverse ST data from several popular platforms and sequencing protocols using a unified representation, facilitating comprehensive analysis as illustrated in Figure 1. In particular, we apply SUICA to enhance the spatial resolution (spatial imputation), to alleviate the drop-out rates (gene imputation), and to smooth the noisy expressions (denoising), respectively. Extensive experimental results demonstrate that SUICA outperforms existing INR variants as well as the state-of-the-art methods in both numerical fidelity and statistical correla-

tion. Moreover, SUICA's imputation ability leads to more cell-type informed clustering results of ST. To summarize, our contributions in this paper are three-fold as follows:

- We introduce SUICA to model ST data as a continuous and compact representation while preserving data authenticity;
- We address the issue that prevent INRs from scaling to the super-high dimensional gene expression of ST by leveraging Graph Autoencoder and a classification-based loss function;
- Extensive experiments show that SUICA achieves superior reconstruction quality and imputation capabilities on various ST datasets, facilitating subsequent analyses.

## 2. Related Work

### 2.1. Implicit Neural Representations

INRs model signals by mapping input coordinates to corresponding signal values using neural networks. Unlike conventional discrete grid-based signal representations, *e.g.*, images, videos, and voxels, INRs are known for continuous modeling, allowing queries at arbitrary locations within the definition domain. Typically implemented as Multi-Layer Perceptrons (MLPs), INRs leverage the smoothness bias of MLPs to provide a certain level of interpolability, allowing effective generalization to unseen coordinates.

The idea of neural networks for function approximation dates back to the 1980s and 1990s (Cybenko, 1989; Hornik, 1991) as universal function approximation theories. From a relatively modern perspective, SIREN (Sitzmann et al., 2020) adopts periodic sine functions as the activation function and learns INRs rich in fine details. Equipped with Random Fourier Features, FFN (Tancik et al., 2020) multiplies the coordinates with a random feature sampled from a normal distribution. BACON (Lindell et al., 2022) further explores the spectral bandwidth of the target signal and progressively regresses in a coarse-to-fine manner.

INRs have been applied across various downstream tasks, including inverse graphics and video compression. NeRF (Mildenhall et al., 2020) parameterizes the 3D scene as an MLP-based radiance field with colors and densities. It applies differential volume rendering to query a series of points in the field and then calculate the weighted sum as the pixel value. Following this parameterization, subsequent works (Liu et al., 2020; Yu et al., 2021; Sun et al., 2022; Hu et al., 2022; Fridovich-Keil et al., 2022; Müller et al., 2022; Peng et al., 2020; Chan et al., 2022; Chen et al., 2022b) further improve the radiance field for efficient training and rendering. Additionally, the continuity of INR is well-suited for application in 3D geometric reconstruction methods (Park et al., 2019; Wang et al., 2021; Yariv

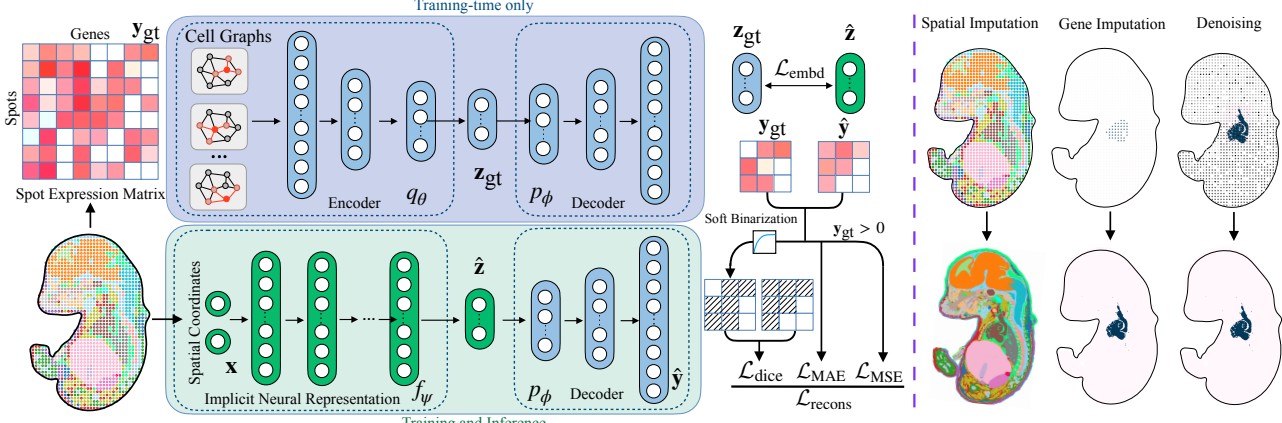

*Figure 2.* The overall pipeline of SUICA. At training-time, a GAE based on cell graphs is trained, with whose pre-trained decoder concatenated to a INR. The INR then maps spot coordinates to the corresponding gene expressions. SUICA is capable of performing spatial imputation, gene imputation, and denoising.

et al., 2021; Oechsle et al., 2021). As for video compression, NeRV (Chen et al., 2021) and subsequent variants (Li et al., 2022; Chen et al., 2023) employ compact INRs to reduce data redundancy. It is noteworthy that existing INR applications map from low-dimensional to low-dimensional spaces, *e.g.*, $\mathbb{R}^2 \to \mathbb{R}^3$ for image regression, and $\mathbb{R}^5 \to \mathbb{R}^4$ for inverse graphics. The use of INRs for mapping from low dimension (*i.e.*, $\mathbb{R}^2$) to super-high dimension (*i.e.*, over 20,000 channels) remains unexplored.

## 2.2. Deep Learning for Spatial Transcriptomics

Deep learning methods are increasingly adopted in ST to enhance spatial resolution, denoise raw data, improve spatial domain clustering, and support more accurate downstream analysis (Zahedi et al., 2024). Given the complexity of capturing cell-to-cell and spatial dependencies, graph-based methods leveraging cell relationships have gained significant traction. Notable examples include GCN-based methods like SpaGCN (Hu et al., 2021), STAGATE (Dong & Zhang, 2022), GraphST (Long et al., 2023), and SEDR (Xu et al., 2024) as well as graph transformer-based model like SiGra (Tang et al., 2023). These graph-based methods progressively refine the modeling of ST data, enabling more nuanced interpretation of complex tissue structure.

ST data often require a higher resolution to accurately capture fine spatial patterns within tissues. Several methods leverage histological images to enhance the spatial resolution of gene expression data, including Convolution Neural Network (CNN)-based models ST-Net (He et al., 2020) and DeepSpaCE (Monjo et al., 2022), and Vision Transformer-based methods such as Hist2ST (Zeng et al., 2022) and TRIPLEX (Chung et al., 2024). In cases without histological images, other approaches (Zhao et al., 2021; 2023; Li et al., 2024) rely solely on spatial and gene expression data to

improve resolution. To generate unmeasured spots profiles, STAGE (Li et al., 2024) trains a spatial location-supervised Autoencoder that enhances resolution by integrating spatial coordinates with gene expression data. In summary, these resolution enhancement methods significantly improve ST quality, improving spatial visualization and facilitating more accurate downstream analyses.

## 3. SUICA

### 3.1. Preliminary

#### 3.1.1. SPATIAL TRANSCRIPTOMICS

ST data can be viewed as an unordered point set, where each point is parameterized by spatial coordinates $\mathbf{x}$. The mRNA readouts are bounded at these sampled locations and represented by a high dimensional vector, where each channel reflects the numerical abundance of a particular mRNA transcript (0 indicates absence or failed capture).

In terms of the spatial distribution of the sampled spots, different ST platforms follow different sampling protocols, *e.g.*, Visium applies a fixed array of probes so the sampled points are distributed uniformly over the whole slice, whereas other common platforms do not guarantee such regularity. Thus, to accommodate general use cases, ST data is often modeled as unstructured without being quantized into regular grids to make CNN-based analysis inapplicable.

#### 3.1.2. CHALLENGES

The key challenge in modeling ST lies in the super-high dimensional representation, often exceeding 20,000 channels. Due to low mRNA capture rates and varying gene expression patterns across cellular states, ST data are typically highly sparse, with zero values comprising up to 90%. A

satisfying model must maintain the inherent sparsity of ST data, as it reflects the true gene expression. However, vanilla INRs are observed to have the tendency to yield normally distributed outputs that are smooth (Lee et al., 2018), rather than the zero-inflated ones of ST, presenting a significant challenge. It is also crucial to ensure the numerical fidelity of non-zero values for accurate cell type identification. It is worth noting that high sparsity complicates the evaluation of reconstruction quality, as an entirely empty prediction may still result in low loss, yet be entirely unacceptable. To this end, the design of the representations and evaluation protocol must address both sparsity and numerical fidelity—an aspect that has, to our knowledge, not been fully explored in the context of INRs.

### 3.2. Method

#### 3.2.1. OVERVIEW

The overall pipeline is illustrated in Figure 2. To construct a compact and dense embedding domain for subsequent INRs, we first incorporate a graph-based encoder and pre-train Graph Autoencoder (GAE) using the given ST slice in a self-regressing manner. With the pre-trained GAE, we obtain the encoded latent representation for all spots, denoted as $\mathbf{z}_{gt}$. We then initiate INR tuning to optimize the neural mapping from ST coordinates to embeddings, targeting $\mathbf{z}_{gt}$. Once the INR reaches a stable state, we attach a decoder that is pre-trained in our GAE. Subsequently, we fix the INR and train the decoder to fit the raw dataset readouts $\mathbf{y}_{gt}$.

Concretely, we perform 3 different tasks with SUICA, namely spatial imputation, gene imputation, and denoising, whose data flows are summarized as follows.

- Spatial Imputation: Say an ST slice $\{(\mathbf{x}, \mathbf{y})\}$, whose data spots are split into a training subset and a test subset. With the training subset, we train SUICA (GAE+INR), and infer the test subset for evaluation.
- Gene Imputation: We randomly mute a part of the gene expressions of the data matrix, fit SUICA with all of the data, and infer all of the $\mathbf{x}$.
- Denoising: The data flow is basically the same as gene imputation, but with injected noise as the degradation.

#### 3.2.2. GRAPH-AUGMENTED AUTOENCODER

The sequenced readouts of ST are known for an extremely skewed distribution, represented as high-dimensional, sparse data. This sparsity exacerbates the curse of dimensionality, rendering data points increasingly dissimilar and challenging to organize efficiently.

To address this, we leverage an AE to transform the high-dimensional raw space into a compact, dense, and informative embedding space. Given the irregular structure, the context information, context or neighborhood information

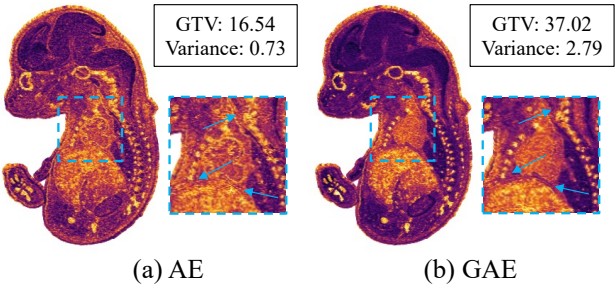

|  |  |
| :---: | :---: |
| (a) AE | (b) GAE |

*Figure 3.* Spectral analysis with the embeddings of AE and GAE. GAE yields structure-aware and disentangled embeddings with high-frequency details. GTV: Graph Total Variation.

is integrated through a graph structure, which we model using a Graph Convolutional Network (GCN). Importantly, the GCN is incorporated solely in the encoder, as the decoder—responsible for reconstructing interpolated embeddings back into the raw space—lacks the requisite graph structure for such integration.

As illustrated in Figure 3, we conduct a graph-based spectral analysis of embeddings generated by both the AE and the GAE. Specifically, we construct a connectivity graph using Euclidean distance between spots ($k = 5$) and calculate the Graph Total Variation (GTV) for embeddings defined on the built graph. Figure 3 is colored based on aggregated edge-wise variations of each vertex, revealing that the GAE produces shaper structural clues. Additionally, we compute the channel-wise variance for both embeddings. Together with the GTV results, these findings demonstrate that GAE offers a more expressive and informative representation, effectively capturing disentangled embeddings for ST.

To train GAE, we adopt the conventional Mean Square Error (MSE) for supervision as

$$\mathcal{L}_{gae} = \frac{1}{|\mathbf{M_y}|} \sum_{\mathbf{M_y}} (\hat{\mathbf{y}} - \mathbf{y}_{gt})^2, \tag{1}$$

where $\mathbf{M_y}$ represents the mask over all of the elements in $\mathbf{y}$ to compute the element-wise average.

#### 3.2.3. EMBEDDING MAPPING

With the pre-trained GAE and encoded representation $\mathbf{z}_{gt}$ in place, we proceed to train the INR to map $\mathbf{x} \rightarrow \mathbf{z}$. The compact, dense embeddings provided by the GAE significantly reduce the workload of the subsequent INR, which must fit an otherwise highly skewed distribution.

For SUICA, we empirically select two baseline architectures: FFN (Tancik et al., 2020) and SIREN (Sitzmann et al., 2020), both of which have demonstrated effectiveness across various data types. In our experiments, we apply SIREN for spatially sparse ST and FFN for denser spatial distributions.

Following conventional INR approaches, we assume

Gaussian-distributed error in the predicted $\hat{\mathbf{z}}$ and use element-wise mean MSE loss for optimization:

$$\mathcal{L}_{\text{embd}} = \frac{1}{|\mathbf{M}_{\mathbf{z}}|} \sum_{\mathbf{M}_{\mathbf{z}}} (\hat{\mathbf{z}} - \mathbf{z}_{\text{gt}})^2. \qquad (2)$$

While $\hat{\mathbf{z}}$ can serve as the mapping output for downstream analysis, we consider it as an intermediate result, decoding this latent code back into the raw space to achieve a more comprehensive reconstruction.

### 3.2.4. DECODING HEAD

A straightforward approach to decoding $\hat{\mathbf{z}}$ back to raw representations is to directly use the pre-trained GAE decoder for end-to-end INR tuning. However, this method encounters two primary challenges. (1) The embedding mapping of $\mathbf{x} \rightarrow \mathbf{z}$ is error-prone, and domain shifts can severely impair decoding performance, compounding with the inherent imperfections in GAE's reconstruction. (2) A pre-trained decoder can become trapped in local minima, impeding the optimization of the embedding mapping if the INR depends on gradients from the decoder. To address (1)(2), we first warm up the INR with Equation (2) alone to stabilize the mapping, then attach the pre-trained decoder to learn the mapping from $\mathbf{z} \rightarrow \mathbf{y}$, leaving the INR fixed. This decoder-only training phase is designed to finetune a case-specific decoder that compensates for mapping errors and minimizes cumulative errors.

In INR regression tasks, norm-based loss functions (*e.g.*, $\ell^2$ norm or MSE) are typically used, as they assume normally (Gaussian) distributed errors. However, for zero-inflated ST data, this assumption is invalid. To handle the imbalanced distribution of zero and non-zero values, we apply Dice Loss (Sudre et al., 2017), which is sensitive to class imbalance and treats the regression task as a quasi-classification one. Dice Loss optimizes for Intersection over Union (IoU) between the prediction map and binary ground truth. Specifically, we use the non-negative half of $\tanh(\cdot)$ to map network outputs into a pseudo-probability range $[0, 1)$, and compute the intersection using element-wise Hadamard products with the ground truth. To avoid division by 0, Dice Loss is computed as:

$$\mathcal{L}_{\text{dice}} = 1 - \frac{2 \sum (\tanh(\hat{\mathbf{y}}) \circ \text{sgn}(\mathbf{y}_{\text{gt}})) + \epsilon}{\sum \tanh(\hat{\mathbf{y}}) + \sum \text{sgn}(\mathbf{y}_{\text{gt}}) + \epsilon}, \qquad (3)$$

where $\text{sgn}(\cdot)$ denotes the sign function returning 0 when the input is 0 and +1/-1 when the input is positive/negative. The full reconstruction loss function is then defined as:

$$\mathcal{L}_{\text{recons}} = \frac{1}{|\mathbf{M}_{\mathbf{y}}^+|} \sum_{\mathbf{M}_{\mathbf{y}}^+} (\hat{\mathbf{y}} - \mathbf{y}_{\text{gt}})^2 + \frac{1}{|\mathbf{M}_{\mathbf{y}}|} \sum_{\mathbf{M}_{\mathbf{y}}} |\hat{\mathbf{y}} - \mathbf{y}_{\text{gt}}| + \lambda \mathcal{L}_{\text{dice}},$$
$$(4)$$

where $\mathbf{M}_{\mathbf{y}}^+$ represents the binary mask for $\mathbf{y}_{\text{gt}} > 0$, and $\lambda$ mitigates Dice Loss's numerical instability.

## 4. Experiments

### 4.1. Datasets & Metrics

For a quantitative benchmarking, we involve a nanoscale resolution Stereo-seq (SpaTial Enhanced REsolution Omics-sequencing) dataset, MOSTA (Chen et al., 2022a). MOSTA consists of a total of 53 sagittal sections from C57BL/6 mouse embryos at 8 progressive stages using Stereo-seq, from which we take 1 slice for each stage (from E9.5 to E16.5) for benchmarking. In addition to Stereo-seq, we also leverage ST data by other common platforms, *i.e.*, Slide-seqV2, 10x Genomics Visium (see Appendix) and MERFISH (see Appendix), to further demonstrate the generalization of SUICA.

As for evaluation of the fitting performance with super-high dimensional data, we focus on 3 aspects, namely numerical fidelity, statistical correlation, and bio-conservation. For numerical fidelity, we apply MSE, MAE (Mean Absolute Error) and cosine similarity to measure the significant, subtle and directional errors between the predicted and ground-truth values respectively. Note that we only measure numerical fidelity on non-zero values considering the zero-inflated distribution of ST. To measure the statistical correlation, we employ Pearson Correlation Coefficient and Spearman's Rank Correlation Coefficient (Spearman's $\rho$), with both of them ranging from -1 to 1. Lastly, to evaluate how well the prediction preserves cellular heterogeneity and spatial coherence within the microenvironment of the slice, we use the Adjusted Rand Index (ARI) as a metric for quantifying bio-conservation leveraging the independent hand-crafted cell type annotations.

### 4.2. Evaluation Protocol

As is the conventional data flow for INRs, SUICA infers gene expressions with coordinates as inputs after fitting the given $(\mathbf{x}, \mathbf{y})$ pairs. We apply SUICA to perform degradation-agnostic reconstruction upon ST data under various common degradations, including spatial sparsity, gene drop-out, and noise, to which we refer as spatial imputation, gene imputation and denoising respectively for clarity. Accordingly, we follow different evaluation protocols: for spatial imputation, we randomly sample 80% of the spots for training, leaving the rest 20% for evaluation; for gene imputation, we randomly mute 70% of the elements in the data matrices; for denoising, a standard Gaussian noise is injected to the raw data. Note that, such reconstruction is completely based on the great approximation power and internal smoothness of INRs without any beforehand knowledge of the degradation type in a reference-free manner.

For comparison, we compare SUICA with rule-based well-known INR variants (FFN (Tancik et al., 2020) and SIREN (Sitzmann et al., 2020)), and the SOTA learning-

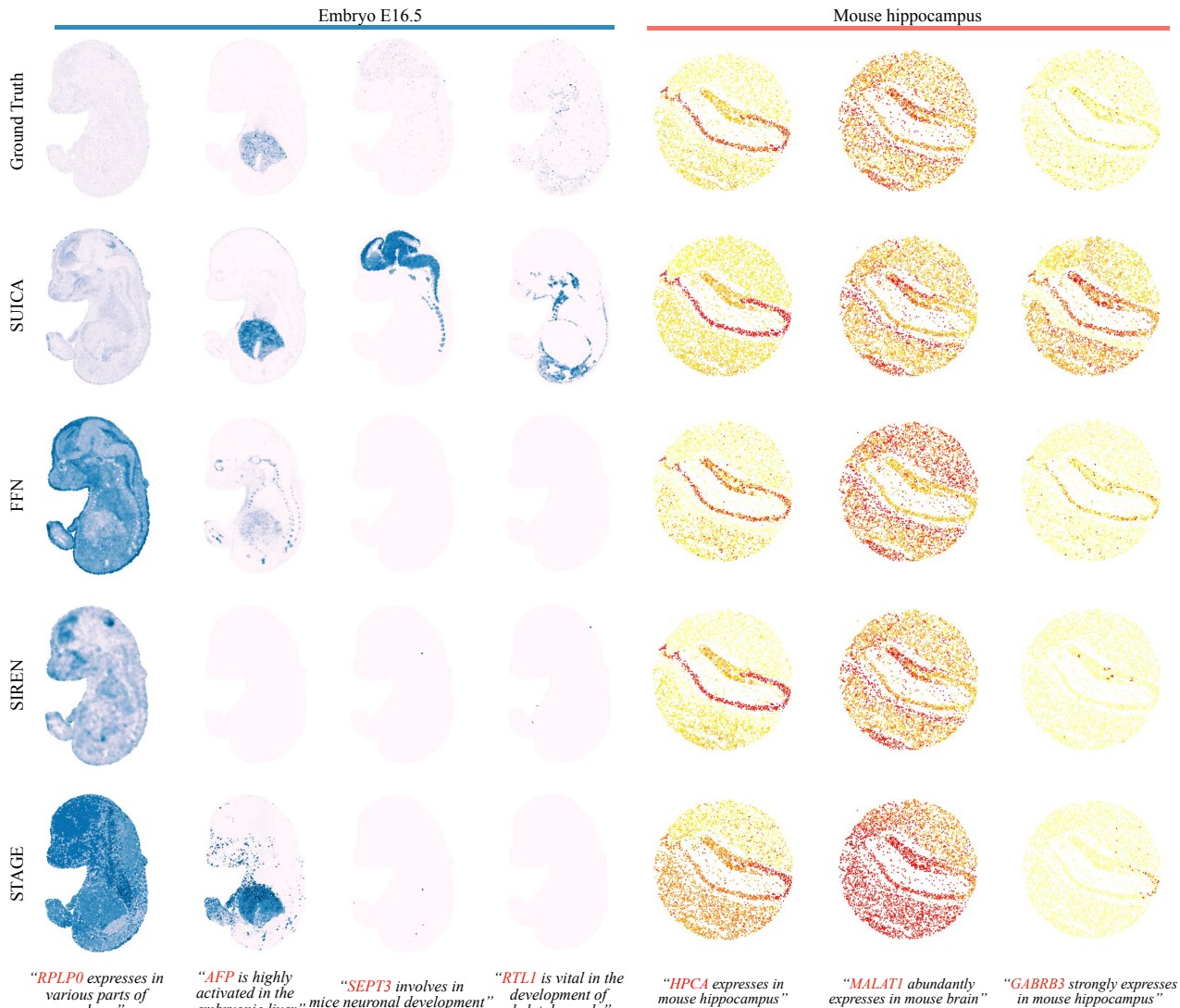

*Figure 4.* Visual comparisons of predicted marker gene expressions on MOSTA dataset (Chen et al., 2022a) and Slide-seqV2 Mouse hippocampus dataset (Stickels et al., 2021), with the descriptions of the markers attached below the results.

based reference-free baseline model STAGE (Li et al., 2024).

### 4.3. Implemented Details

We implement SUICA with PyTorch. To construct the $k$-NN graph for GCN, we set $k = 5$, including the given cell itself. During the pre-training phase of GAE, we use Adam with a learning rate of $1e^{-5}$ for 200 epochs. After obtaining low-dimensional cell embeddings, we train the INR with Adam at a learning rate of $1e^{-4}$ for $1k$ epochs to learn the embedding mapping. Subsequently, the INR is frozen, with the pre-trained decoder trained for an additional $1k$ epochs using Adam with the same learning rate. STAGE (Li et al., 2024) is trained till convergence for benchmarking. All experiments can be conducted on 1 NVIDIA RTX 4090.

### 4.4. Spatial Imputation

**Stereo-seq MOSTA** SUICA outperforms INR methods (Tancik et al., 2020; Sitzmann et al., 2020) and STAGE (Li et al., 2024) in predicting the gene expression of unseen spots in the MOSTA dataset (Chen et al., 2022a) as presented in Table 1. SUICA not only achieves the best MAE, MSE and cosine similarity, but also achieves higher correlation. Besides, SUICA surprisingly manages to strengthen the biological signals, even 3.9% above the ARI calculated with the ground truth.

Figure 4 showcases SUICA's ability to closely predict ground-truth gene expression levels, *e.g.*, for *RPLP0* gene that highly involves in the ribosomal function and consistently expresses across various regions of mouse em-

*Table 1.* Quantitative benchmarking results of spatial imputation on MOSTA dataset (Chen et al., 2022a) and Slide-seqV2 Mouse hippocampus dataset (Stickels et al., 2021). **Bold** figures are best scores and underlined figures are second-best. The respective reference ARI scores are 0.312 (Stereo-seq MOSTA) and 0.182 (Mouse hippocampus Slide-seqV2). MAE/MSE: $\times 10^{-2}$ for Stereo-seq MOSTA.

| Methods | Stereo-seq MOSTA | | | | | | Mouse hippocampus Slide-seqV2 | | | | | |
|---|---|---|---|---|---|---|---|---|---|---|---|---|
| | MAE↓ | MSE↓ | Cosine↑ | Pearson↑ | Spearman↑ | ARI↑ | MAE↓ | MSE↓ | Cosine↑ | Pearson↑ | Spearman↑ | ARI↑ |
| FFN (Tancik et al., 2020) | 6.51 | 1.20 | 0.706 | 0.718 | 0.400 | 0.143 | 0.378 | 0.215 | 0.499 | 0.442 | 0.274 | 0.0523 |
| SIREN (Sitzmann et al., 2020) | 7.21 | 1.31 | 0.661 | 0.678 | 0.247 | 0.289 | 0.383 | 0.216 | 0.494 | 0.452 | 0.248 | 0.110 |
| STAGE (Li et al., 2024) | 6.52 | 1.11 | 0.732 | 0.747 | 0.365 | 0.139 | 0.351 | 0.198 | 0.587 | **0.483** | **0.314** | 0.0361 |
| **SUICA** (Ours) | **5.66** | **0.85** | **0.797** | **0.792** | **0.447** | **0.343** | **0.265** | **0.125** | **0.752** | 0.473 | 0.308 | **0.111** |

bryo (Taylor & Pikó, 1992; Ozadam et al., 2023), SUICA accurately predicts this uniform expression pattern, while other methods overemphasize it in specific regions. SUICA also accurately localizes *AFP* expression to the liver in the mouse embryo (Kwon et al., 2006). Beyond accurately predicting gene expression in unobserved regions, SUICA is also capable for imputation, enhancing the data to better reflect true underlying biological signatures. For example, despite its low expression level in the ground-truth, SUICA successfully imputes *SEPT3*, a gene involved in neuronal development, effectively restoring this signal in the brain region. These results highlight SUICA's capacity not only to interpolate but also to enrich underlying biological signatures, making it a useful tool for imputing and enhancing spatial gene expression data with high fidelity preserved.

**Mouse hippocampus Slide-seqV2** Slide-seqV2 allows to sequence RNAs with a spatial resolution of 10 $mm$. The Mouse hippocampus dataset (Stickels et al., 2021) is applied to evaluate the effectiveness of SUICA. As shown in Table 1, SUICA achieves a substantially lower MAE and a notably higher cosine similarity (marking a 16.5% improvement) compared to other methods. Figure 4 illustrates the ground-truth and predicted gene expression from the benchmarking methods. Like other methods, SUICA accurately predicts the expression of hippocampus marker genes, such as *HPCA* (Hippocalcin) (Park et al., 2017) and *MALAT1*, a gene abundantly expressed in the mouse brain (Park et al., 2017). SUICA also demonstrates its ability to impute the underlying biological signals, as evidenced by *GABRB3*, which is consistently and strongly expressed in the mouse hippocampus (Tanaka et al., 2012). These findings suggest that SUICA provides reliable predictions that closely reflect ground-truth data, capturing key gene expression patterns in the Slide-seqV2 platforms.

### 4.5. Bio-conservation Analysis of Predicted Cell Type Clusters

Biological variance conservation (Bio-conservation) refers to the degree to which a computational method or model preserves biologically meaningful features, such as cell types, gene expression patterns, or cellular relationships, relative

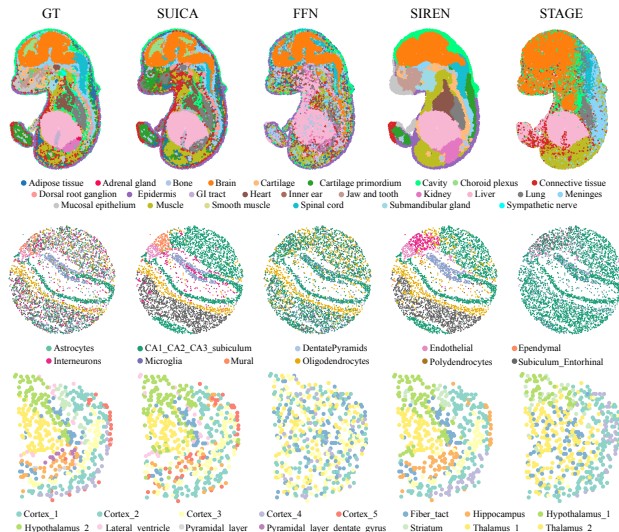

*Figure 5.* Spatially visualized comparison on bio-conservations of predicted spots on MOSTA mouse embryo E16.5, Slide-seqV2 mouse hippocampus, and Visium-Mouse brain.

to the ground-truth data (Luecken et al., 2022). A high bio-conservation score indicates that the predicted data aligns closely with biological reality, capturing key aspects of cell identity, functional states, or tissue structure that are essential for accurate analysis.

In addition to quantitative bio-conservation metrics like ARI, Figure 5 provides a visual assessment of bio-conservation by coloring predicted spots according to dominant cell types in individual clusters. For a fair comparison, we ensured that the number of clusters matched the number of cell-type labels in the dataset.

In E16.5, apart from the highest ARI among benchmarking methods, SUICA also offers the closest spatial visualization of cell types to the ground truth. Compared to other methods, SUICA captures fine-grained structures, such as the choroid plexus in the brain, the gastrointestinal (GI) tract, and the muscle cells surrounding the heart. FFN's (Tancik et al., 2020) predictions appear chaotic, with most cell-type predictions misaligned with the ground-truth labels. While SIREN (Sitzmann et al., 2020) correctly captures general cell types patterns, it overlooks many details detected by

*Table 2.* Quantitative benchmarking results of gene imputation and denoising on MOSTA dataset (Chen et al., 2022a). **Bold** figures are best scores and underlined figures are second-best. MAE/MSE: $\times 10^{-2}$.

| Methods | Gene Imputation | | | | | Denoising | | | | |
|---|---|---|---|---|---|---|---|---|---|---|
| | MAE↓ | MSE↓ | Cosine↑ | Pearson↑ | Spearman↑ | MAE↓ | MSE↓ | Cosine↑ | Pearson↑ | Spearman↑ |
| FFN (Tancik et al., 2020) | 4.88 | 0.963 | 0.731 | 0.610 | 0.251 | 7.90 | 1.95 | 0.266 | 0.285 | 0.0523 |
| SIREN (Sitzmann et al., 2020) | 6.44 | 1.12 | 0.675 | 0.652 | 0.124 | 7.91 | 1.97 | 0.112 | 0.103 | 0.0166 |
| STAGE (Li et al., 2024) | 4.69 | 0.738 | **0.802** | 0.705 | 0.264 | 7.60 | 1.66 | 0.606 | 0.630 | 0.182 |
| **SUICA** (Ours) | **4.30** | **0.724** | 0.798 | **0.714** | **0.317** | **6.03** | **0.934** | **0.733** | **0.737** | **0.379** |

SUICA, such as the connective tissue around the cartilage primordium in the mouse's foot region and the cavity surrounding the heart. STAGE (Li et al., 2024) identifies several major cell types in the mouse embryo—such as liver and brain—but misses smaller, yet important, cell populations.

In the Mouse hippocampus dataset, SUICA also demonstrates its ability to capture both global and fine-grained features, accurately identifying endothelial cells, Ependymal cells, and inter-neurons. In contrast, FFN and STAGE fail to detect smaller cell populations, while SIREN misclassifies inter-neurons. Similarly, on the Visium-Mouse Brain dataset, SUICA uniquely identifies specific cell populations, such as Lateral Ventricle cells and Cortex_5, which are absent from the predictions of other methods.

### 4.6. Gene Imputation & Denoising

Since INRs are known for its internal smoothness, SUICA is able to perform channel-wise gene imputation and denoising upon muted or contaminated gene expressions. In this way, SUICA can be seen as a reference-free degradation-agnostic restoration method for ST.

We benchmark the quantitative results in Table 2 upon Stereo-seq MOSTA (Chen et al., 2022a). Note that the pipeline is exactly identical except for the degradation pattern. SUICA manages to handle the contaminated inputs in almost all metrics. It also shows that GAE manages to make it easier for INRs to model the smoothness in the super-high dimensional space of gene expressions.

### 4.7. Ablation Study

We showcase two examples to illustrate how the proposed modules in SUICA contribute to the final model performance, namely the E16.5 embryo of Stereo-seq MOSTA (spatially dense, 121,756 cells, 26,159 genes, with FFN (Tancik et al., 2020) as backbone) and the Visium Human Brain (spatially sparse, 4,910 cells, 36,592 genes, with SIREN (Sitzmann et al., 2020) as backbone), under the setting of spatial imputation.

As can be indicated from Table 3, the spatial density and sequencing depth may influence SUICA's effectiveness. For

*Table 3.* Ablation study on model design of SUICA. MSE: $\times 10^{-2}$ for Embryo E16.5 while $\times 10^{-3}$ for Human Brain.

| Settings | Embryo E16.5 | | | Human Brain | | |
|---|---|---|---|---|---|---|
| | MSE↓ | Cosine↑ | Pearson↑ | MSE↓ | Cosine↑ | Pearson↑ |
| Vanilla INR | 2.35 | 0.668 | 0.653 | 9.33 | 0.756 | 0.747 |
| +AE | 1.60 | 0.789 | 0.751 | 11.27 | 0.695 | 0.691 |
| +Dice | 1.48 | 0.806 | 0.747 | 7.05 | 0.826 | 0.800 |
| +Graph | 1.47 | 0.807 | 0.761 | 5.67 | 0.860 | 0.846 |

the Embryo E16.5 dataset, the primary performance improvement comes from the AE and the quasi-classification loss. In contrast, in the Human Brain dataset, adding AE causes a performance decrease, while the GCN has a more significant impact on overall performance. Intuitively, we attribute this difference to the varying spatial sparsity of different ST techniques and GAE alleviates such issue by incorporating spatial context.

## 5. Conclusion

In this paper, we seek to model the super-high dimensional and sparse nature of ST data, enhancing both spatial resolution and gene expression with the smooth prior inherent in INRs. To this end, we present SUICA, a powerful INR variant tailored to model ST in a continuous and compact manner. Using a Graph Autoencoder, SUICA maps zero-inflated raw data into a lower-dimensional embedding space, preserving high-frequency details and making the complex embedding mapping more feasible for INRs. The INR fitted embeddings are then decoded to the raw expression with the decoding head preventing the prediction error from accumulating. To encourage the sparsity in the final predictions, we leverage a quasi-classification loss term as a regularizer, preserving both the sparsity and numerical fidelity of non-zero values. Extensive experiments on Stereo-seq, 10x Genomics Visium, and Slide-seqV2 datasets demonstrate SUICA's effectiveness, yielding improvements in both in-silico metrics and biologically meaningful analyses, with enhanced spatial resolution and biological signatures. It is firmly believed by us that SUICA will be an inspiring attempt from both INRs and ST perspectives. We encourage readers to check the appendix for complementary insights and extended evaluations.

## Acknowledgements

This work was supported in part by Three Lakes Foundation, the Canadian Institutes of Health Research (CIHR) [PJT-180505], the Fonds de recherche du Québec - Santé (FRQS) [295298, 295299], the Meakins-Christie Chair in Respiratory Research, AMED [JP23tm0524003], JSPS KAKENHI [24KK0209, 24K22318, 22H00529], and JST-Mirai Program [JPMJMI23G1].

## Impact Statement

SUICA offers a comprehensive solution to the challenges of spatial sparsity, gene drop-out, and noise in spatial transcriptomics data analysis. It is compatible with a wide range of sequencing platforms, making it broadly accessible to research teams. Given the high cost of obtaining ST data, SUICA empowers the scientific community to enhance data quality and accelerate the discovery of novel biological insights.

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

## A. Background

Spatial Transcriptomics (ST) is a technique that representing the gene expression at individual spots within a slice of sample, providing a detailed picture of cellular activity and composition in the micro-environment. ST sequencing is typically performed on fresh frozen tissue samples and is broadly categorized into two main types: sequencing-based ST and imaging-based ST. Despite differences in methodology, both technologies deliver complementary measurements of *in situ* gene expression. The sequencing-based ST technologies, including Stereo-seq, 10x Visium, Slide-seqV2, usually rely on spatially indexed surfaces to encode spatial information (Figure 6). These methods employ different spatial indexing strategies: 10x Visium uses a microarrayer spotting robot, these spots are 50–200 $\mu m$ in size, to deliver a unique barcode to a fixed, known location on the surface of a slide (Tian et al., 2023); Slide-seq applies 10 $\mu m$ diameter nanobeads for spatial barcoding (Tian et al., 2023); Stereo-seq uses DNA nanoball-patterned arrays for *in situ* RNA capture to create high-resolution spatial transcriptomics sequencing (Chen et al., 2022a). Following barcoding, mRNA diffuses to spatially indexed primers, undergoes reverse transcription, cDNA amplification, and short-read sequencing. This process generates transcript reads paired with barcode sequences, localizing each detected transcript to specific pixels.

For the imaging based-ST methods, RNA molecules are specifically tagged with fluorescent probes by complementary hybridization. These probes are then imaged using fluorescence microscopy. MERFISH, for instance, maps RNA molecules with binary barcodes encoded with error-correcting codes to ensure accuracy. Sequential rounds of fluorescence imaging detect the presence or absence of fluorescence, decoding the RNA identity and precisely mapping molecules to their spatial locations.

We also want to mention the fact that ST data are not always delivered jointly with histology images as a reference. For example, Slide-seq, DBiT-seq, MERFISH offer high-resolution spatial gene expression data but lack corresponding histological images. Even in cases where histology is available, challenges in precise alignment between histological features and ST data can arise due to tissue deformation, sectioning artifacts, or registration inaccuracies. These limitations highlight the importance of designing reference-free models that do not rely on histological guidance, making them more broadly applicable and robust across different ST platforms and experimental conditions.

## B. Model Details & Insights

**FFN vs. SIREN** As mentioned in the main body, it is observed that when applied to ST with low spatial density,

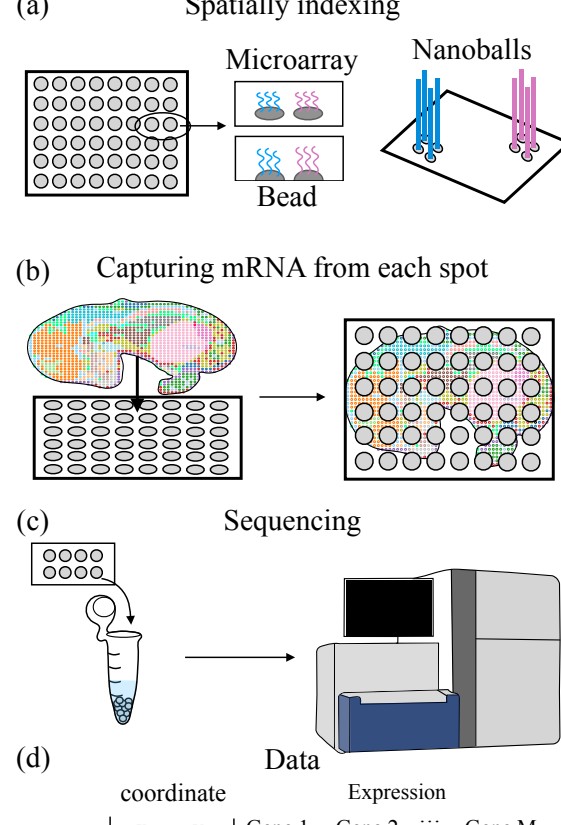

*Figure 6.* Workflow of sequencing-based spatial transcriptomics sequencing. (a) Spatially indexing, with microarrayer for 10x Visium, nanobead technology for Slide-seqV2 and nanoball-based strategy for Stereo-seq; (b) Placing the slice of tissue on the indexed surface and capturing mRNA from each spot; (c) Collecting spots and conducting sequencing; (d) The generated data matrix with gene expression and location information for each spot.

SIREN (Sitzmann et al., 2020) outperforms FFN (Tancik et al., 2020) as the backbone of SUICA, while for high-resolution ST slices, FFN prevails. To better showcase such phenomenon, we provide two representative cases of the mouse embryos E9.5 and E16.5 in Table 4. We globally set the angular velocity $\omega$ of the periodical activation function in SIREN (Sitzmann et al., 2020) as 30 which is adopted by default and the mapping size of the Random Fourier Features as 256. For more information for these two magic numbers, please refer to the respective papers. Note that we do not intend to showcase any superiority among INR variants. Since we adopt a modular design that is agnostic to specific INR implementations, we are able to take advantage of different implementations to meet varying requirements.

*Table 4.* Quantitative results of SUICA empowered by different INR backbones. MSE: $\times 10^{-2}$.

| Backbones | Embryo E9.5 | | | Embryo E16.5 | | |
|---|---|---|---|---|---|---|
| | MSE↓ | Cosine↑ | Pearson↑ | MSE↓ | Cosine↑ | Pearson↑ |
| FFN (Tancik et al., 2020) | 0.55 | 0.752 | 0.799 | 1.47 | 0.807 | 0.761 |
| SIREN (Sitzmann et al., 2020) | 0.52 | 0.766 | 0.814 | 1.62 | 0.786 | 0.745 |

*Table 5.* Quantitative comparison on Stereo-seq MOSTA dataset (Chen et al., 2022a) between SUICA and NGP (Müller et al., 2022). MAE/MSE: $\times 10^{-2}$.

| Methods | MAE↓ | MSE↓ | Cosine↑ | Pearson↑ | Spearman↑ |
|---|---|---|---|---|---|
| NGP (Müller et al., 2022) | 6.39 | 1.14 | 0.729 | 0.742 | 0.415 |
| SUICA | 5.66 | 0.85 | 0.797 | 0.792 | 0.447 |

**Explicit Alternatives** Recently, researchers are employing explicit or hybrid alternatives (Müller et al., 2022; Chen et al., 2022b) to replace the fully implicit representations (Tancik et al., 2020; Sitzmann et al., 2020; Lindell et al., 2022) for a faster inference speed. For a more comprehensive benchmarking, we here compare SUICA with the trainable hash-based encoding (Müller et al., 2022) (denoted as NGP) on Stereo-seq MOSTA dataset (Chen et al., 2022a). The quantitative results are shown in Table 5. For the implementation, we re-compile the library of `tiny-cuda-nn` to support `float32`, to align with other methods.

**CNN** It is easy to notice that the network architecture in SUICA is almost fully MLP-based except for the GCN. Such architecture does not explicitly involve any local inductive bias that CNNs usually do. The underlying reason for doing so is that along the sequencing depth of ST, it is assumed that the channels (*i.e.*, specific gene segments) are permutation-invariant even though there might be some statistical correlations. In this way, it is not supposed to rely on the context information along the sequencing depth, but for the correlation among cells, we model such context by the GCN.

**PCA** In SUICA, we reduce the dimension of raw gene expressions from thousands to 32 by the encoder of GAE, which is recovered later by the decoder. The function of the GAE could be replaced by PCA (Principal Component Analysis), by a pre-fitted linear projection. However, it is obviously challenging for the linear transformation of PCA and inverse PCA to model the complex distribution of ST, leading to a great loss in the reconstructed signal. Empirically, it is observed that when encoding and decoding with PCA and inverse PCA, the IoU between the predicted zero maps and the ground truth decreases to 0, indicating a complete failure of recovering the sparsity, while SUICA maintains an IoU of around 0.9.

**VAE** Variational Autoencoders (VAEs) are a common alternative to AEs, designed to generate samples from a prob-abilistic latent space, typically modeled as Gaussian distributions. While VAEs offer generative capabilities through variational inference and learn smoother, more continuous latent spaces compared to AEs, they often produce blurry outputs (Zhao et al., 2017) and struggle to match AEs in reconstruction accuracy (Dai et al., 2020). These issues might lead to ambiguity in gene expression that is not wanted. These limitations stem from the inherent trade-off between the quality of latent representation learning and reconstruction fidelity in VAEs. Although variants like beta-VAE (Higgins et al., 2017) allow for adjustable weighting between these objectives, the issue persists. In our setting, where numerical fidelity and reconstruction accuracy take precedence over generative and latent representation quality, we select AE over VAE for SUICA's design, leveraging its ability to accurately reconstruct high-dimensional raw gene expression data.

**Reconstruction Loss** In SUICA, we train the INR and the decoder separately to minimize the accumulated prediction error, on which we would like to further offer some insights. Obviously, a pre-trained decoder is necessary; otherwise the decoder would become a trivial extension of the INR. It is also easy to understand that such decoder requires finetuning to bridge the gap between the pseudo ground-truth embeddings $\mathbf{z}_{gt}$ and the predicted embeddings $\hat{\mathbf{z}}$. The problem that prevents this two-stage training strategy from merging into one end-to-end stage is that when using $\mathcal{L}_{recons}$ through the decoder to supervise the INR, the loss will become unstable. We attribute this observation to the optimizing objectives of $\mathcal{L}_{embd}$ and $\mathcal{L}_{recons}$ are somehow contradictory. Though this problem can be alleviated by carefully adjusting the training-relevant hyper-parameters, we opt to adopt the two-stage training strategy for simplicity and stable performance.

# C. STAGE

In the main text, we involve STAGE (Li et al., 2024) for both quantitative and qualitative comparisons as a baseline for histology-free spatial imputation (densification) of gene expressions. In addition to solely showing the superiority of SUICA, we would like to shed some light on this matter.

Regarding the benchmarking task of imputation, we identify two critical sub-tasks, namely the reconstruction of gene expression and the spatial mapping. STAGE models imputation as a generative task, where an AE is adopted to self-regress the raw representations while enforcing the latent as the corresponding 2D coordinates. In this way, the two sub-tasks are coupled as one. Considering the challenges posed, the closely coupled paradigm may not lead to satisfactory results. On the contrary in SUICA, the two sub-tasks are well decoupled, where the spatial mapping is performed by the INR while the description as well as the

*Table 6.* Quantitative benchmarking results of spatial imputation on two Visium ST cases (Palla et al., 2022; Wei et al., 2022). Note that for Visium-Human Brain, there is no annotation of cell type for the evaluation of ARI, while reference ARI is 0.428 for Mouse Brain. **Bold** figures are best scores and underlined figures are second-best. MAE/MSE: $\times 10^{-2}$ for Human Brain while $\times 10^{-1}$ for Mouse Brain.

| Methods | *Visium-Human Brain* | | | | | *Visium-Mouse Brain* | | | | | |
|---|---|---|---|---|---|---|---|---|---|---|---|
| | MAE↓ | MSE↓ | Cosine↑ | Pearson↑ | Spearman↑ | MAE↓ | MSE↓ | Cosine↑ | Pearson↑ | Spearman↑ | ARI↑ |
| FFN (Tancik et al., 2020) | 5.76 | 0.881 | 0.772 | 0.786 | 0.402 | 5.95 | 5.85 | 0.832 | 0.741 | 0.581 | 0.000587 |
| SIREN (Sitzmann et al., 2020) | 6.58 | 0.933 | 0.756 | 0.747 | 0.196 | 5.35 | 4.29 | 0.878 | 0.804 | 0.647 | 0.359 |
| STAGE (Li et al., 2024) | 6.19 | 0.805 | 0.795 | 0.772 | 0.223 | 4.55 | 3.20 | 0.918 | **0.825** | **0.666** | 0.140 |
| TRIPLEX (Chung et al., 2024) | **4.75** | **0.560** | **0.881** | **0.850** | 0.319 | 9.35 | 14.0 | 0.00 | -0.00682 | -0.00715 | 0.358 |
| UNIv2 (Chen et al., 2024) | 7.30 | 1.41 | 0.723 | 0.633 | 0.129 | 6.94 | 7.88 | 0.790 | 0.631 | 0.425 | 0.228 |
| **SUICA** (Ours) | 4.99 | 0.567 | 0.860 | 0.846 | **0.445** | **3.68** | **2.45** | **0.932** | 0.800 | 0.660 | **0.393** |

reconstruction is taken care of the GAE. The progressive training paradigm also guarantees that each module is doing the assigned job, keeping the coupling at minimum.

According to the experimental results, SUICA not only achieves better quantitative scores for benchmarking, but also exhibits much richer bio-conservation for downstream applications.

## D. Human and Mouse Brains Visium

Different from Stereo-seq datasets, Visium is a lower-resolution but more affordable ST technology, on which we additionally compare the model performance. The experiments are conducted under the task of spatial imputation. Due to the lack of independent cell-type annotations, ARI is not applicable in Visium-Human Brain dataset (Wei et al., 2022). On the Visium-Mouse Brain dataset (Palla et al., 2022), SUICA demonstrates improved ARI, which indicates that the predictions can more accurately describe the underlying cellular heterogeneity. Note that for readers' information, we also involve SOTA histology-aided imputation methods TRIPLEX (Chung et al., 2024) and UNI (Chen et al., 2024) for comparison, which additionally has access to extra reference. We have also found that TRIPLEX is rather sensitive to dataset-specific characteristics.

## E. MERFISH

MERFISH is an imaging-based ST technique that enables the highly multiplexed imaging of RNA molecules in cells while maintaining their spatial context. Compared to the Slide-seqV2, 10x Visium, and Stereo-seq technologies used in our manuscript, MERFISH data can quantify significantly fewer genes for individual cells. In Table 7, we evaluate the performance of SUICA using a human heart MERFISH dataset (Farah et al., 2024) with 228,635 cells and 238 genes, with the setting of spatial imputation. SUICA achieves a significant lower mean absolute error (MAE) and mean square error (MSE), and the cosine similarity is at least 2.28% higher than other methods, showing its high numerical fidelity. Following the bio-conservation analysis scheme, we

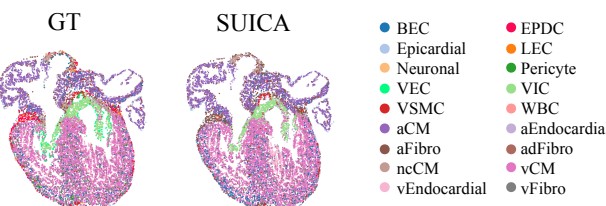

*Figure 7.* Spatially visualized predicted spots on MERFISH human heart (Farah et al., 2024).

*Table 7.* Quantitative results of spatial imputation on MERFISH human heart (Farah et al., 2024). MAE/MSE: $\times 10^{-1}$.

| Methods | MAE↓ | MSE↓ | Cosine↑ | Pearson↑ | Spearman↑ |
|---|---|---|---|---|---|
| FFN (Tancik et al., 2020) | 6.06 | 5.86 | 0.840 | 0.717 | 0.554 |
| SIREN (Sitzmann et al., 2020) | 5.54 | 4.98 | 0.864 | 0.759 | 0.606 |
| STAGE (Li et al., 2024) | 5.48 | 5.10 | 0.870 | 0.709 | **0.558** |
| SUICA | **4.65** | **3.92** | **0.892** | **0.718** | 0.548 |

annotate the cell types for predicted spots of MERFISH data using dominant cell types in individual clusters. We visualize the spatial bio-conservation by coloring the cell types in Figure 7. The results show that the SUICA predictions mimic the ground-truth evidenced the capability of SUICA to predict unseen spots in the MERFISH datasets.

## F. Ablation on Resolution

To help with the data-efficiency analysis of SUICA, we perform experiments under the task of spatial imputation with E16.5 embryo of Stereo-seq MOSTA (Chen et al., 2022a), the results of which are shown in Table 8. The percentage refers to the the proportion of training samples with regard to the whole dataset, while the test set remains the same (20% of all spots).

## G. Data Pre-processing

To enable the evaluation of Pearson Correlation Coefficient and Spearman's Rank Correlation Coefficient, before the experiments we remove empty rows and lines to make sure each spot is at least expressed by a limited number of genes.

Note that the outputs of conventional INR-based tasks are

*Table 8.* Ablation study on the data-efficiency of SUICA. % represents the proportion of the spots used for training while test set remains 20%. MAE/MSE: $\times 10^{-2}$.

| % | MAE↓ | MSE↓ | Cosine↑ | Pearson↑ |
|---|---|---|---|---|
| 80% | 8.01 | 1.47 | 0.807 | 0.761 |
| 60% | 7.96 | 1.52 | 0.801 | 0.752 |
| 40% | 8.00 | 1.59 | 0.790 | 0.739 |
| 20% | 8.14 | 1.62 | 0.786 | 0.738 |

strictly bounded, where the linear layer is usually clamped or compressed by sigmoid. For ST data, we do not have such assumption and only employ ReLU as the final activation. Therefore, as a common practice in ST analysis (Melsted et al., 2021; Hwang et al., 2018), we remove the genes whose expressions are overly high and normalize each cell by total counts over all genes of the cell so that every cell has the same total count after normalization while keeping the original sparsity.

## H. Limitations

One of the significant limitations in SUICA roots in the case-by-case training paradigm of INRs. Similar to NeRF (Mildenhall et al., 2020), for each case, *i.e.*, each ST slice, we need to training a new model from scratch. Considering the overwhelming heterogeneity of different ST data, we temporarily compromise in this issue but acknowledge that incorporating domain knowledge to make SUICA generalizable is an interesting future work.

Another potential limitation is that SUICA prefers high-quality ST data, in terms of both spatial density and sequencing depth, as can be indicated from the experimental results we provide. SUICA exhibits the most performance gain in Stereo-seq compared to other platforms and when the gene expressions no longer meet the super-high dimensional sparse assumption, SUICA will degenerate to a vanilla INR, as is consistent with the results of MERFISH in Table 7.

