# OpenReview forum: "SUICA: Learning Super-high Dimensional Sparse Implicit Neural Representations for Spatial Transcriptomics"
_ICML.cc/2025/Conference — ICML 2025 poster_

### Official Review · Reviewer_RB2n · 2025-02-21

**Overall Recommendation:** 3

**Summary:**

This paper proposed a method for jointly denoising and improving resolution for spatial transcriptomics data.

## update after rebuttal

I raised score.

**Claims And Evidence:**

Yes, their claim is supported.

**Essential References Not Discussed:**

Yes, I think they need to include more baselines. Please see the questons.

**Experimental Designs Or Analyses:**

Yes, I have checked the soundness.

**Methods And Evaluation Criteria:**

I have a question about problem definition, which is discussed in the question section.

**Other Comments Or Suggestions:**

The presentaiton looks good.

**Other Strengths And Weaknesses:**

I have raised several questions in the Question section.

**Questions For Authors:**

This paper is very interesting, but I have several questions about problem definition, model implementation, and evaluations. I will consider raising my score if the authors fully address my concerns.

1. I am confused about the imputation function. From my understanding, imputation in ST data means predicting unmeasured genes, which always need external resources, such as reference scRNA-seq data, to predict those unseen ones. Could you please explain how to use your pipeline to impute gene expression? Otherwise, is it the same as denoising?

2. What is the difference between z_gt and z^hat in their implementation, could the authors visualize their latent representations and discuss potential differences? I think we can directly use a GNN to encode spatial transcriptomic data with the awareness of spatial context, what is the drawback of this design?

3. The ablation studies seem different from their presentations, as their listed loss functions include L_dice,  L_mae, and L_mse, but in section 4.7, the presented results are different.

4. The authors perform experiments based on both ST data with different resolutions, such as Slide-seq V2 and Visium. It will be interesting to investigate the performance differences across data with different resolutions. Could the authors elaborate on this point?

5. The cited link of STAGATE seems wrong, as the correct one should be: https://www.nature.com/articles/s41467-022-29439-6. I think it is also interesting to include a comparison with recent baselines such as SEDE (https://genomemedicine.biomedcentral.com/articles/10.1186/s13073-024-01283-x)

**Relation To Broader Scientific Literature:**

Yes, their contirbution is important for computational biology.

**Theoretical Claims:**

Yes, they look good.

---

> ### Author Rebuttal · Authors · 2025-03-30
>
> We want to thank Reviewer RB2n for the valuable comments and suggestions, which have greatly helped us improve the quality of our work.
>
> 1. **Definition of imputation**: Thank you for pointing out this issue. In the manuscript, we use "spatial imputation" to refer to predict gene expressions at 2D locations not included in the training subset. This setting can also be seen as "super resolution" or "densification". As for the task of "gene imputation", we randomly mute a part of gene expressions and apply SUICA to do the reconstruction, so that such muted values could be restored. This setting is more relevant to "reconstruction". Please refer to our response to Reviewer C5Qu for the detailed data flow.
>
> 2. **Latent code**: $z_\textrm{gt}$ refers to the encoded latent code by the GAE's encoder; $\hat{z}$ represents the fitted $z$ by the INR. The goal of our model is to map spatial coordinates of spots to their corresponding gene expressions. However, as shown in the ablation study (Table 3), a vanilla INR can not perform well due to the dimensional challenge of ST. Therefore, we enforce INRs to learn the mapping between spatial coordinates and lower dimensional embeddings $z_\textrm{gt}$. Ideally, $\hat{z}$ should accurately align with $z_\textrm{gt}$ which allows GAE’s decoder to seamlessly restore the gene expressions. We have visualized the embeddings of $\hat{z}$ and $z_\textrm{gt}$ using UMAPs. The results show that the INR can effectively reproduce the embeddings learned by GAE’s encoder, allowing SUICA to infer the embeddings of arbitrary locations for spatial imputation. Please kindly refer to [this anonymous link](https://imgur.com/a/RlYePtD) for the visualization.
>
> 3. **Why not use GNN only**: We agree with you that GNN can encode ST data. However, we
>     - apply GAE's encoder to gain low dimensional embeddings to enhance INR (Table 3),
>     - employ INRs to map spatial coordinates of arbitrary spots to their embeddings,
>     - and use GAE's decoder to restore the gene expression from embeddings.
>
>      Without these key factors, using GNN only is difficult to map spatial coordinates to gene expression.
>
>
> 4. **Detailed ablation on loss**: For the reviewer's information, we break down the loss terms leveraged in SUICA and offer a more detailed ablation on the identical data in Table 3. To better show the loss combinations' effect on the sparsity, we additionally report IoU of non-zero areas for reference. We make our choice as MSE+MAE+Dice to ensure a relatively stable trade-off between numerical fidelity and statistical correlation.
>
>    | Embryo E16.5        | MSE $\times 10^{-2}$$\downarrow$ | Cosine$\uparrow$ | Pearson$\uparrow$ | IoU$\uparrow$   |
>    | ------------ | --------- | ------ | ------- | ----- |
>    | MSE          | 1.23      | 0.843  | 0.522   | 0.558 |
>    | MSE+Dice     | 1.22      | 0.843  | 0.572   | 0.674 |
>    | MSE+MAE      | 1.60      | 0.789  | 0.751   | 0.937 |
>    | MSE+MAE+Dice | 1.48      | 0.806  | 0.747   | 0.928 |
>
>    | Human Brain  | MSE $\times 10^{-3}$$\downarrow$ | Cosine$\uparrow$ | Pearson$\uparrow$ | IoU$\uparrow$   |
>    | ------------ | --------- | ------ | ------- | ----- |
>    | MSE          | 10.50     | 0.723  | 0.507   | 0.532 |
>    | MSE+Dice     | 10.72     | 0.719  | 0.574   | 0.696 |
>    | MSE+MAE      | 11.27     | 0.695  | 0.691   | 0.933 |
>    | MSE+MAE+Dice | 7.05      | 0.826  | 0.800   | 0.925 |
>
> 5. **Resolution**: We agree with Reviewer RB2n that the performance is relevant to the resolution of data. However, it is somehow difficult to ablate such influence because in practice, the resolution is usually coupled with other factors that will have a potential effect on the performance. For instance, a high spatial resolution usually leads to a high drop-out rate, which is also challenging for modeling. We have performed experiments with different held-out rates in the response to Reviewer Q3xk, and we hope the results of such artificial simulation could be of some insights.
>
> 6. **Comparison with SEDR**: As is suggested, we benchmark against SEDR. Since SEDR can not perform spatial imputation, we report its gene imputation and denoising performance. However, when attempting to let SEDR reconstruct the raw gene expressions of *Stereo-seq MOSTA* (the default use case is to adopt PCA-reduced expressions), OOM is observed with RTX 4090, so we instead report the scores on *Visium-Human Brain*. MAE/MSE: $\times 10^{-2}$.
>    | Gene Imputation | MAE$\downarrow$  | MSE$\downarrow$   | Cosine$\uparrow$ | Pearson$\uparrow$ | Spearman$\uparrow$ |
>    | - | - | - | - | - | - |
>    | STAGE | 7.13 | 1.69  | 0.692  | 0.667   | 0.110    |
>    | SEDR | 37.0 | 91.4  | 0.0    | 0.048   | 0.076    |
>    | SUICA | 5.64 | 0.699 | 0.835  | 0.829   | 0.391    |
>    | **Denoising** |  | | | | |
>    | STAGE | 5.49 | 0.757 | 0.814  | 0.740   | 0.140    |
>    | SEDR | 47.5 | 90.8  | 0.0255 | 0.117   | 0.0793   |
>    | SUICA| 4.20 | 0.582 | 0.860  | 0.751   | 0.201    |

---

> > ### Comment · Reviewer_RB2n · 2025-04-01
> >
> > Thank you for your comments. I will keep my score as I am already in the side of acceptance. Good luck!

---

> > > ### Author Response · Authors · 2025-04-07
> > >
> > > Thank you very much for your prompt acknowledgement and response! Should you have further questions, we will be glad to participate in the discussion.

---

### Official Review · Reviewer_Q3xk · 2025-03-11

**Overall Recommendation:** 3

**Summary:**

The paper proposes SUICA, a method for continuous modeling of spatial transcriptomics data by combining a graph-augmented autoencoder (GAE) with implicit neural representations (INRs). The key idea involves compressing high-dimensional, sparse gene expression data into a low-dimensional latent space using a GAE, then mapping spatial coordinates to this latent representation via an INR (using either FFN or SIREN), and finally decoding back to the full gene expression space. To address sparsity in spatial transcriptomics, the authors reformulate the reconstruction loss with a quasi-classification approach based on Dice Loss. Experimental evaluations on multiple datasets (including Stereo-seq MOSTA, Slide-seqV2 mouse hippocampus, and Visium-Mouse brain) demonstrate improvements in spatial and gene imputation as well as bio conservation.

**Claims And Evidence:**

- The authors claim that SUICA outperforms conventional INR variants and existing methods (e.g., STAGE) in terms of numerical fidelity, statistical correlation, and biological conservation. Experimental results are presented over several datasets with metrics such as gene MAE, MSE, cosine similarity, correlations, and ARI of cell type. SUICA achieves new SOTA in most cases.
- However, while the authors claim their approach uniquely handles super-high dimensionality, the model input to GAE is actually dimensionality-reduced input via PCA  (Appendix B, line 699). This critical detail is omitted from the main text, and Figure 2 is extremely misleading as it depicts identical input and output sizes for the GAE.

**Essential References Not Discussed:**

N/A

**Experimental Designs Or Analyses:**

Given that the GAE input is PCA-transformed, the paper would benefit from a detailed ablation study on PCA's contribution to the results.

**Methods And Evaluation Criteria:**

The use of GAE is interesting, but it's unclear whether performance improvements stem from integrating expression from neighboring spots to reduce noise, which is an already well-established approach in the field. For example the original MOSTA paper uses bin 50 (summing 50 spots) for downstream analysis.

**Other Comments Or Suggestions:**

N/A

**Other Strengths And Weaknesses:**

N/A

**Questions For Authors:**

- Is the regression target of FFN or SIREN baseline also PCA transformed?
- How is data preprocessing conducted, including normalization, qc's.
- Given the extremely complex model with three training stages, how were hyperparameters selected?
- Can you please share your thoughts on previous sections?

**Relation To Broader Scientific Literature:**

- The paper adequately discusses relevant works in the field.
- The work presents an interesting use of INR in spatial biology, which is novel to the best of review's knowledge.

**Theoretical Claims:**

N/A

---

> ### Author Rebuttal · Authors · 2025-03-30
>
> We appreciate the efforts of Reviewer Q3xk by offering comments and suggestions, which would definitely help us to improve the manuscript for its potential readers. We are also afraid that there is some critical misunderstanding towards SUICA, so we hope this rebuttal could lead to a better alignment and kindly ask Reviewer Q3xk to re-evaluate the technical contribution of SUICA.
>
> 1. **SUICA does not use PCA and operates in full gene space**: We apologize for the confusion. To clarify, the pipeline of SUICA does **NOT** use PCA for dimensionality reduction at any stage and both the input and output of the GAE are in full-gene space. During test time, it predicts gene expression profiles of the same dimensionality as the raw input, using only spatial coordinates.
>
>     The discussion about PCA in Appendix B is intended to solve the potential concern of whether the encoder and decoder of GAE could be replaced by PCA and iPCA. However, this is not part of SUICA’s pipeline. The dimension reduction and reconstruction in SUICA are respectively conducted by the encoder and decoder of GAE. Therefore, the notion that input to GAE is dimensionality-reduced input via PCA is **incorrect**. We apologize again for being unclear and will make revisions to Section 4.3 and Appendix B to prevent further misunderstanding.
>
> 2. **The use of GAE**: The effectiveness of GAE has been discussed and verified in Section 3.2.2 and Section 4.7. We agree with the reviewer's intuitive idea that the root of performance is by leveraging neighborhood information and exploiting the correlation among data. But we would like to emphasize that the INR in SUICA also plays an important role in the continuous modeling of ST, making it possible to query the gene expressions at arbitrary locations, instead of performing denoising. To this end, we believe that the technical contributions of SUICA are non-trivial.
>
> 3. **Regression target for SIREN & FFN**: The regression target for SIREN and FFN is the raw gene expressions without being processed by PCA, as is also the target of SUICA.
>
> 4. **Data preprocessing**: As is mentioned in Appendix F, the preprocessing includes cells and gene filtering and count depth scaling. We first filtered spots with less than 200 genes expressed and genes that were detected in less than 3 spots. Then we removed spots whose total number of raw counts is lower than the threshold (set as 200 empirically). Then we performed count depth scaling (``sc.pp.normalize_total()``) following the tutorial of scanpy.
>
> 5. **Training-relevant hyperparameters**: SUICA applies staged training to obtain satisfactory results with super-high dimension data, which to our knowledge is acceptable and far from being extremely complex. The training-relevant hyperparameters are set according to empirical results, with which SUICA is able to perform stably across varying ST platforms. It is worth noting that, we use the identical set of hyperparameters for ST data from different platforms.

---

> > ### Comment · Reviewer_Q3xk · 2025-04-05
> >
> > Thanks for your detailed rebuttal! I have increased my score.

---

> > > ### Author Response · Authors · 2025-04-07
> > >
> > > Thank you for looking into the rebuttal! Should you have further questions, we will be glad to participate in the discussion.

---

### Official Review · Reviewer_c5Qu · 2025-03-13

**Overall Recommendation:** 4

**Summary:**

The authors introduce SUICA, an implicit neural representation-based ST prediction method, which demonstrates another way of gene/spatial imputation for ST. This provides a different take on ST inference frameworks, which predominantly were based on image-based or neighborhood-cell based without relying on nearby coordinates.

**Claims And Evidence:**

I think the authors provide claims that are always backed up by evidence.

**Essential References Not Discussed:**

There are few key references on "super-resolution" of ST that I felt were not discussed, mostly based on image-based algorithms.

[1] Hu, Jian, et al. "Deciphering tumor ecosystems at super resolution from spatial transcriptomics with TESLA." Cell systems 14.5 (2023): 404-417.
[2] Zhang, Daiwei, et al. "Inferring super-resolution tissue architecture by integrating spatial transcriptomics with histology." Nature biotechnology 42.9 (2024): 1372-1377.

**Experimental Designs Or Analyses:**

Yes.

**Methods And Evaluation Criteria:**

Yes

**Other Comments Or Suggestions:**

See above

**Other Strengths And Weaknesses:**

I think this is a timely contribution to the field, which has been dominated by image-based prediction or purely cell-based GCN-based approach. However, I think a bit more efforts are required to explain/convince readers who are predominantly familiar with current methods and not so with INR-based. If the authors are able to address few additional suggestions, I am willing to increase the score.
- Figure 2 is great for explaining the training pipeline, but I think there needs to be an illustrative description (also corresponding detailed explanation in the text) on how the test-time inference is carried out. What are the required inputs? is it just the subset of coordinates? Is it just the subset of the cell-level expressions? Can these be randomly distributed (as in the experiments)? This will help the readers understand the situation in which SUICA is available or not
- There are a few super-resolution ST approaches, although they are based on image-based inputs (TESLA, ISTAR). I would like to see comparison with them if possible (TESLA should be easier to implement, ISTAR maybe not given the rebuttal timeframe)
- Ideally I want to see a bit more ablation in the evaluation. The authors experiment with 80% randomly sampled spots for training and remaining 20% for evaluation. What if it's on the lower-end of the training? Such ablation experiments will help readers understand amount of data required to ensure "decent" imputation performance.
- Can the authors comment on whether SUICA needs to be trained everytime for each ST plane? Based on my understanding, it does seem that SUICA needs to be fitted for each plane. However, for the embryo dataset, where it seems to consist of serial sections of ST, maybe only few sections can be used for training and the other purely for evaluation? In other words, I would love to see generalization performance/argument for SUICA.
- It would be great to see a bit more comparison with image-based ST imputation. TRIPLEX is great, but it uses underpowered vision encoders that might not be as great for downstream imputation performance. Maybe the authors can reference some baselines from recent HEST study [1]?

References
[1] Jaume, Guillaume, et al. "Hest-1k: A dataset for spatial transcriptomics and histology image analysis." Advances in Neural Information Processing Systems 37 (2024): 53798-53833.

**Questions For Authors:**

See above

**Relation To Broader Scientific Literature:**

I think this reflects a good timely contribution to the field.

**Theoretical Claims:**

No theoretical claims provided

---

> ### Author Rebuttal · Authors · 2025-03-30
>
> We are sincerely thankful for Reviewer C5Qu's comments and suggestions, and we hope the following responses could help to resolve the concerns.
>
> 1. **Pipeline of SUICA**: Thank you for raising the issue about the confusion of data flow. We will modify the manuscript to help readers better understand our data flow. Here, we summarize the data flow according to Section 4.2 of the manuscript into the following points:
>
>    - Spatial imputation: Take an ST slice {($x$,$y$)} as an example, whose data spots are split into a training subset {($x_\textrm{train}$,$y_\textrm{train}$)} and a test subset {($x_\textrm{test}$,$y_\textrm{test}$)}. With the training subset, we train the self-regressing GAE, whose encoder encodes all $y_\textrm{train}$ as $z_\textrm{train}$. Then we start fitting the INR to approximate the mapping from $x_\textrm{train}$ to $y_\textrm{train}$ (with intermediate supervision of $z_\textrm{train}$). For test-time inference, SUICA **only needs $x_\textrm{test}$** to infer corresponding $y_\textrm{test}$.
>    - Gene imputation: We randomly mute a part of the gene expressions of the data matrix, fit SUICA with all of the data, and infer **all of the $x$**. We expect the prediction $\hat{y}$ to be imputed.
>    - Denoising: The overall data flow is basically the same as gene imputation, but with injected noise as the degradation.
>
>    As a result, **the required input at test time is only the coordinates** and we do not make any assumption of the spatial distribution (e.g., a uniform distribution as regular grids), which means SUICA is able to be both trained and tested with unstructured coordinates. We will follow the reviewer's suggestion to elaborate this issue more clearly in the revised manuscript.
>
> 2. **Image-based super-resolution**: Thank you for suggesting benchmarking against other image-based super-resolution methods. We additionally compare SUICA with TESLA and UNIv2[1] with *Visium-Human Brain* and *Visium-Mouse Brain*. Note that SUICA is reference-free and does not incorporate any image information. We also report IoU of non-zero areas for reference to emphasize the predicted sparsity. MAE/MSE: $\times 10^{-1}$.
>    | Human Brain | MAE$\downarrow$ | MSE$\downarrow$ | Cosine$\uparrow$ | Pearson$\uparrow$ | Spearman$\uparrow$ | IoU$\uparrow$ |
>    | - | - | - | - | - | - | - |
>    | TESLA | 0.280 | 0.00533 | 0.857 | 0.828 | 0.421 | 0.859 |
>    | UNIv2 | 0.730 | 0.141 | 0.723 | 0.633 | 0.129 | 0.388 |
>    | SUICA | 0.498 | 0.0567 | 0.860 | 0.846 | 0.445 | 0.931 |
>    | **Mouse Brain** | | | | | | |
>    | TESLA | 4.00 | 2.58 | 0.877 | 0.786 | 0.619 | 0.619 |
>    | UNIv2 | 6.94 | 7.88 | 0.790 | 0.631 | 0.425 | 0.0 |
>    | SUICA | 3.68 | 2.45 | 0.932 | 0.800 | 0.660 | 0.655 |
>
> 3. **Held-out rates**: Thank you for emphasizing the necessity of ablation study with different training-test ratios. Under the setting of spatial imputation (or "super resolution"), we evaluate the performance of SUICA with different held-out rates (the proportion of spots kept for test) with *Embryo E16.5*. According to the results, we find there is no significant performance drop when the held-out rate raises. MAE/MSE: $\times 10^{-2}$.
>    | Held-out Rates | MAE$\downarrow$  | MSE$\downarrow$  | Cosine$\uparrow$ | Pearson$\uparrow$ |
>    | - | - | - | - | - |
>    | 20% | 8.01 | 1.47 | 0.807 | 0.761 |
>    | 40% | 7.98 | 1.54 | 0.799 | 0.748 |
>    | 60% | 7.94 | 1.52 | 0.802 | 0.751 |
>    | 80% | 8.13 | 1.66 | 0.781 | 0.735 |
>
> 4. **Generalizability**: As is discussed in the limitations of Appendix G (Line 770), INRs are case-by-case optimized. At the current stage, we consider this limitation not fatal since the data heterogeneity across different ST platforms prevents the construction of a unified dataset and many SOTA methods also adopt the case-by-case paradigm, e.g, STAGE and SEDR. But we do acknowledge that incorporating INRs for modeling ST in a generalizable way is a promising direction, and there have been some preliminary attempts for images and 3D assets [2].
>
> 5. **Baselines from HEST**: Thank you for suggesting the benchmarking against more image-based imputation methods. Here we have selected UNIv2 [1] and please refer to the previous table for the results.
>
> [1] Chen, Richard J., et al. "Towards a general-purpose foundation model for computational pathology." *Nature Medicine* 30.3 (2024): 850-862.
>
> [2] Ma, Qi, et al. "Implicit Zoo: A Large-Scale Dataset of Neural Implicit Functions for 2D Images and 3D Scenes." *The Thirty-eight Conference on Neural Information Processing Systems Datasets and Benchmarks Track*.

---

> > ### Comment · Reviewer_c5Qu · 2025-04-02
> >
> > Thank you for putting the rebuttal together. I still have few more questions based on the rebuttal
> > 1. **Super-resolution** I wouldn't say TESLA is image-based, since it is not predicting ST from the image. Could authors comment on why TESLA, which does not require complicated learning procedure, seems to be quite competitive with SUICA?
> > 2. **Held-out** I actually meant to have the same test set (not increasing like you presented) and change the size of the training set. This will help us analyze SUICA's data-efficiency property, as I expect larger training dataset size to yield better performa

---

> > > ### Author Response · Authors · 2025-04-07
> > >
> > > Thank you very much for the prompt response towards our rebuttal. We hope the follow-up response could resolve the concerns.
> > >
> > > 1. **Competitive performance of TESLA**: We mainly attribute the competitive performance of TESLA to the histology used as inputs. It is supposed to be easier for a network to exploit the piece-wise smoothness with pixels than with the raw gene expressions alone. For instance, the super-high dimensionality makes it difficult to measure the expression-wise distances between spots. Besides, we conduct additional experiments under the task of gene imputation to better reveal the performance with degraded inputs, where we find that TESLA is quite sensitive in this case.
> > >
> > >    |   |       | MSE$\downarrow$ | Cosine$\uparrow$ | Pearson$\uparrow$ |
> > >    | --------------- | ----- | --------------- | ---------------- | ----------------- |
> > >    | *Human Brain*   | TESLA | 0.00863         | 0.662            | 0.534             |
> > >    |                 | SUICA | 0.00699         | 0.835            | 0.829             |
> > >    | *Mouse Brain*   | TESLA | 0.788           | 0.668            | 0.499             |
> > >    |                 | SUICA | 0.607           | 0.825            | 0.690             |
> > >
> > > 2. **Data-efficiency of SUICA**: We augment new experiments according to the suggestion. The percentage refers to the the proportion of training samples with regard to the whole dataset, while the test set remains the same. MAE/MSE: $\times 10^{-2}$.
> > >
> > >    | % training | MAE$\downarrow$ | MSE$\downarrow$ | Cosine$\uparrow$ | Pearson$\uparrow$ |
> > >    | ---------- | --------------- | --------------- | ---------------- | ----------------- |
> > >    | 80%        | 8.01            | 1.47            | 0.807            | 0.761             |
> > >    | 60%        | 7.96            | 1.52            | 0.801            | 0.752             |
> > >    | 40%        | 8.00            | 1.59            | 0.790            | 0.739             |
> > >    | 20%        | 8.14            | 1.62            | 0.786            | 0.738             |

---

### Decision · Program_Chairs · 2025-05-01

**Decision:**

Accept (poster)

**Comment:**

This paper proposes SUICA that models spatial transcriptomics data in a continuous and compact manner by using Implicit Neural Representations (INRs). The reviewers had concerns regarding missing key references, unclear explanation of test-time inference, limited evaluations scope, lack of baselines. The authors addressed most of the concerns, and the strengths of the paper outweighs the drawbacks.